**communications** engineering

# A microphysiological system for studying barrier health of live tissues in real time
Ryan Way[1], Hayley Templeton[2], Daniel Ball[1], Ming-Hao Cheng [1], Stuart A. Tobet[2,3] & Thomas Chen [1,3] ✉

Epithelial cells create barriers that protect many different components in the body from their external environment. Increased gut barrier permeability (leaky gut) has been linked to several chronic inflammatory diseases. Understanding the cause of leaky gut and effective interventions are elusive due to the lack of tools that maintain tissue's physiological environment while elucidating cellular functions under various stimuli ex vivo. Here we present a microphysiological system that records real-time barrier permeability of mouse colon in a physiological environment over extended durations. The system includes a microfluidic chamber; media composition that preserves microbiome and creates necessary oxygen gradients across the barrier; and integrated sensor electrodes for acquiring transepithelial electrical resistance (TEER). Our results demonstrate that the system can maintain tissue viability for up to 72 h. The TEER sensors can distinguish levels of barrier permeability when treated with collagenase and low pH media and detect different thickness in the tissue explant.

Barrier tissues composed of epithelial monolayers are the body's first line of defense for protecting a mammalian host from its external environment[1]. These barriers help maintain cellular and tissue homeostasis[1–3]. Disruption of the intestinal barrier has been implicated in pathologies including inflammatory bowel disease (IBD)[4], neurodegenerative disorders[5,6], and numerous others[7–9]. Consequently, there is a demand for measuring the integrity of cellular barriers, prompting researchers to turn to techniques such as Transepithelial/transendothelial electrical resistance (TEER). TEER measurements represent the electrical resistance of the cellular barrier in real time and is a widely accepted measure of barrier integrity[3]. Traditionally, TEER has been used on cell monolayers, devoid of the complex three dimensional and heterogenous cellular environment of barrier tissues in vivo. Thus, there is a need for a microphysiological system that maintains the cellular complexity of the in vivo barrier tissue while maintaining the ability to measure barrier breakdown with TEER over the course of a full experiment.

Classically, epithelial barrier integrity is interrogated via fluorescently labeled probes to visualize the paracellular flux across the epithelial layer[10–15]. However, the importance of real-time data on barrier integrity during an experiment has led to an increased desire for using TEER to examine epithelial barrier properties. A widely used device for TEER measurements is the Transwell[2,16], where a cell monolayer is grown inside of a well-plate and TEER is measured using two "chopstick" electrodes. Transwells are desirable for their simplicity, cost, and high throughput. However, Transwells are not suited for in-depth investigations of cellular barrier function because of the simplified in vitro model without media flow and high variability from chopstick electrodes. To bolster the functionality and physiological representation of in vitro models, organ-on-a-chip (OoC) devices have been employed. These are more advanced devices containing engineered organ tissues and incorporating microfluidics to ensure continuous supply of fresh media containing vital nutrients and accounting for physical force faced by epithelia such as cyclic strain, fluid sheer stress, and mechanical stretching[11,17]. The results from the experiments using both Transwell and OoC devices have enhanced our understanding of the transepithelial transport process, however, they do not replicate the cellular complexity of the intact tissue barrier in vivo[2,8,13,14,18].

To create a better physiological model some researchers have moved towards ex vivo tissue explant devices. The Ussing chamber was one of the first and most commonly used tools to assess epithelial permeability in ex vivo experiments by measuring paracellular flux using electrical measurements. The Ussing chamber was originally designed to understand the phenomenon of active NaCl transport across frog skin[19,20]. This technique set the groundwork and created the first model for epithelial ion permeability. More modern designs of Ussing systems have accommodated many different types of epithelial barriers found inside animals. These barriers include skin, intestinal, esophageal, and more[3,20–23]. Common shortcomings of current Ussing chamber designs include the use of expensive benchtop equipment; static media within the chamber; low throughput; and bulky experimental setup. Static media makes it difficult to maintain viability of live ex vivo tissue for an extended period of time inside of devices. With reduced tissue viability, experiments using live tissues in an Ussing chamber have often been limited to under 3 h[20,24]. Due to the limited viability and lack

[1]Department of Electrical & Computer Engineering, Colorado State University, Fort Collins, CO, USA. [2]Department of Biomedical Sciences, Colorado State University, Fort Collins, CO, USA. [3]School of Biomedical Engineering, Colorado State University, Fort Collins, CO, USA. ✉e-mail: thomas.chen@colostate.edu

of tissue equilibrium, there are limited interrogations that can be performed on the tissue and long-term affects cannot be addressed.

The intestinal barrier as live tissue is coordinated across complex biomaterials (e.g., mucus) and cellular elements that range from bacteria in the intestinal lumen to multiple epithelial cell types on the surface of the gut wall and cell types inside the wall that include, immune, stromal, neural, and muscle cells. Ex vivo models maintain these complex interactions providing a much more realistic physiological model than cell culture, epithelial monolayers, or OoC. One crucial distinction between cell culture and live tissue models lies in the variety of cell types and biological structures present within the model[14]. The most common intestinal in vitro models are comprised of immortalized Caco-2 cells that originated from a human intestinal tumor[3]. These cells are useful for absorption studies and are capable of forming tight junctions between cells in the monolayer, but lack many of the other structures that intestinal epithelium have[3,20]. There have also been OoC devices with intestinal organoid models cultured from stem cells[8,14]. Intestinal organoid models can add a layer of epithelial cell diversity in the formation of intestinal villi and crypts, but still lack important cell components such as the immune, neuronal, or muscle systems present in vivo[14].

The ex vivo model is capable of providing cellular diversity closest to that in vivo, however, the challenge of maintaining the tissues viability over the course of an experiment has hindered the effectiveness of many current devices. Enhancing tissue viability is essential in crafting ex vivo models that mimic physiological conditions more accurately. This enables researchers to extend experimentation durations, thereby capitalizing on the benefits of enhanced cellular diversity across various experimental setups. Advancements in microfluidic device design and appropriate media composition have allowed live tissue inside a microphysiological chamber to maintain cellular heterogeneity and other key health markers over a long-term experiment (>48 h)[12,14,25,26]. Microfluidics create a more accurate physical microenvironment for the live tissue explant (e.g., fluid flow and shear stress) and the media composition provides the tissue with appropriate nutrient diffusion and oxygen gradients[14]. Existing devices capable of long-term explant viability are limited to data observed after the tissue is removed from the device, 48 h or more after it was dissected from the animal[12,25]. Incorporating TEER measurements inside such microphysiological devices increase the temporal resolution of changes in barrier function that might be caused by external perturbations (e.g., infection or chemical irritation). Without real-time TEER measurements in ex vivo models, tissue would need to be taken and removed from the experiment setup at multiple time points to assess barrier integrity. This would result in the use of more animals to obtain enough tissue to perform long time point experiments (i.e., 72 h) and would greatly increase labor and time to achieve results.

This paper presents a microphysiological system for ex vivo tissues capable of longer-term viability and real-time TEER measurement with integrated sensors and electronics. The difficulties of using ex vivo tissue (e.g., leaking, intestinal wall damage, etc.,) has limited the number of systems capable of integrating these two crucial attributes. However, incorporation of ex vivo tissue in TEER measurements is necessary to create a model that most accurately represents in vivo conditions. If key components of a barrier system are missing in drug development, the drug may not perform as well in vivo. With most of the complex interactions maintained in the biological model and the ability to measure TEER throughout an experiment, this system is equipped to investigate essential biological questions and hypotheses for drug discovery and developmental research when compared to in-vitro models.

Overall, our contribution is threefold: first, the microphysiological system is capable of keeping intestinal tissue explants viable for up to 72 h by utilizing physiologically relevant media and microfluidic channels. Within the media, there is an oxygen gradient, a prebiotic to maintain the hosts natural microbiome, and a normoglycemic environment. Second, the TEER measurement of the ex vivo tissue is supported by integrated electrodes and backend electronics to record real-time TEER measurements throughout the entirety of the experiment at a user-specified interval, thus, allowing

users to observe the evolution of barrier integrity in real time. Third, the system architecture is scalable allowing multiple microfluidic chambers to be connected to the system at a time, allowing for multiple controlled experiments using live tissues from the same donor. The proposed device is designed for intestinal tissue slices extracted from a mouse. However, the chamber design can be easily adapted to other barrier tissues.

## Methods

### Microfluidic chamber design

The microfluidic chamber (Fig. 1a–e), to be referred to as chamber for short, was designed using CAD (Autodesk, Inc) and fabricated using Anycubics UV sensitive resin and SLA printer. To avoid harmful effects from uncured resin, each chamber is fully cured using UV light and thoroughly rinsed with isopropyl alcohol. It is further sterilized in a low-temperature autoclave. The chamber consists of two halves assembled to hold the tissue and make connections to the integrated TEER electrodes. Each half chamber is composed of the following (Fig. 1a): the chamber body; two 1 mm thick PDMS layers for holding the electrode chip in place; one gold electrode chip; an aluminum clamp; and a printed circuit board (PCB) with spring headers that connect to the electrode chip. The fully assembled chamber (Fig. 1b) consists of two halves, the top half has spikes to hold the tissue tight when the chamber is sealed; the bottom half has corresponding openings for these spikes and is where the tissue is placed before closing the chamber. Figure 1c shows how the tissue explant positioned on the bottom half chamber, before and after the experiment. When the tissue is enclosed in the device, the chamber spikes puncture around the outer edge of the tissue but leaves the center untouched. A PDMS layer is placed over the spikes of the top chamber to create a flush seal against the tissue and prevent leaks between the chamber halves after the chamber is closed. A fully assembled chamber is shown in Fig. 1d, e in the front and the back view of the device, respectively.

Tubing is connected to each chamber half through Luer lock connectors. Media is pumped into the chamber using custom-designed syringe pumps where users can specify start and stop time points throughout the experiment period. With the tissue positioned over the circular opening between the chamber halves, a barrier is formed between the two media flows, one for the serosal side and the other for the luminal side of the tissue. The microfluidic path flows over the opening on each side exposing the tissue to the media composition. Providing balanced flow for the tissue inside the chamber is important for controlling shear stress and extending tissue viability[10,13,20]. The chamber was designed using 3D fluid simulations (CFD, Autodesk Inc) to make the media flow over the tissue area with uniform velocity as possible. Figure 1f shows an example of the flow simulations performed during design.

Each chamber half has its own PCB breakout board that connects to the glass chip electrodes through gold spring headers. The spring headers are compressed against the chip during assembly. The PCB on the top half chamber has external wire connectors for connecting to the bottom halve PCB. The bottom PCB includes a card edge connector that is plugged into the top of the enclosure for the microphysiological system (see section "System Overview below"). The chamber, when plugged into the system, is oriented vertically, making the media flow in from the bottom and out above the tissue (Fig. 1f). This orientation helps push air bubbles to the top and get them pushed out of the media outlet during experiments. Air bubbles can injure the tissue and cause large deviations for the TEER measurement.

### System overview

The entire electronic support system is housed in a metal enclosure (Fig. 1g, h) to shield all electronics from the external physical environment as well as EMF noise. USB ports provide user configuration and control of the experiment from the host computer. The connectors on the microfluidic chambers and the system enclosure are all universal, allowing for plug-and-play functionality. The current implementation can hold up to three chambers at a time (Fig. 1g, h). The entire system has a footprint of a typical laptop computer designed to allow it to fit into a limited environment chamber space during experiments. The supporting electronics are

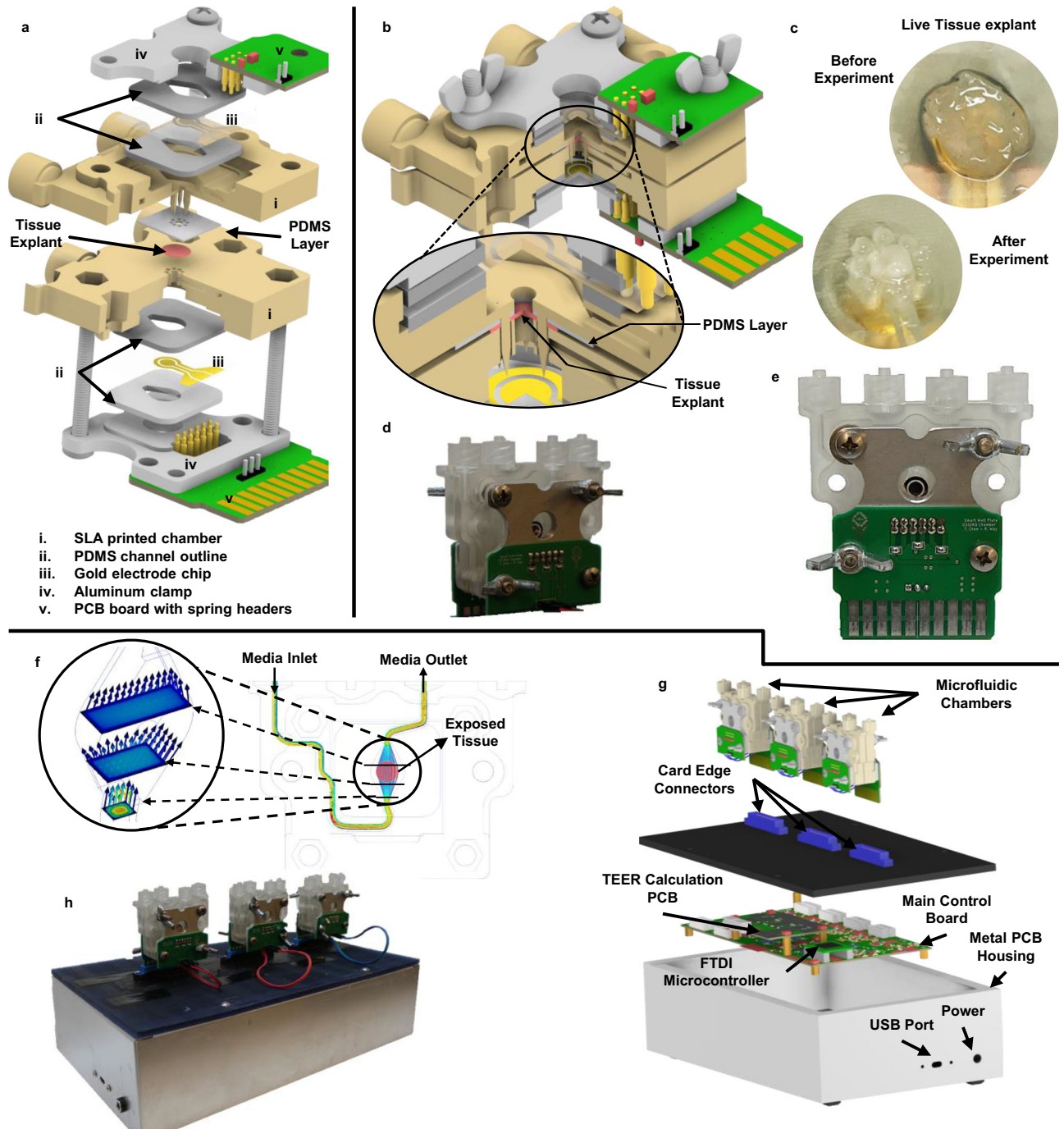

i.    SLA printed chamber
ii.   PDMS channel outline
iii.  Gold electrode chip
iv.   Aluminum clamp
v.    PCB board with spring headers

**Fig. 1 | Major components of the microphysiological system.** Including the microfluidic chamber (**a–f**) and the system-level housing (**g–h**). **a** Expanded view of a full chamber, with all components labeled. **b** Closed chamber with a closeup view of the tissue and PDMS clamped between two chamber halves. **c** Tissue explant before and after the experiment. **d, e** The actual manufactured chamber assembled (front and back, respectively). **f** Flow simulation through the chamber's microfluidics. **g** Expanded view of the microphysiological system. **h** The manufactured and fully assembled system with three chambers connected the system.

responsible for signal acquisition, signal conditioning and amplification, analog to digital conversion, and communication with the host computer via the USB protocol. A custom-built graphic user interface (GUI) was designed to allow user configurations and real-time control for the experiment. The TEER measurement data are acquired by the host computer and TEER results are calculated and displayed in the GUI.

## Electronic circuits for TEER measurement
The block diagram of the electronic circuits to perform TEER measurement is shown in Fig. 2a. The TEER measurement is operated in the constant-current mode to avoid accidental over-current to damage electrodes and tissues inside the chamber. An FTDI module is used to provide interface between the host computer and the on-board electronics using the USB protocol. This interface allows users to control all internal enable signals from the GUI. A signal generator is used to generate a user defined AC voltage signal. This voltage signal is used to control a Howland current source (HCS), creating an AC current stimulus signal to the TEER electrodes. The HCS (Fig. 2b) is chosen because it can easily achieve high output impedance, signal-to-noise ratio (SNR), and is fully programmable through external control voltages[27,28]. Up to three parallel current stimulation signals

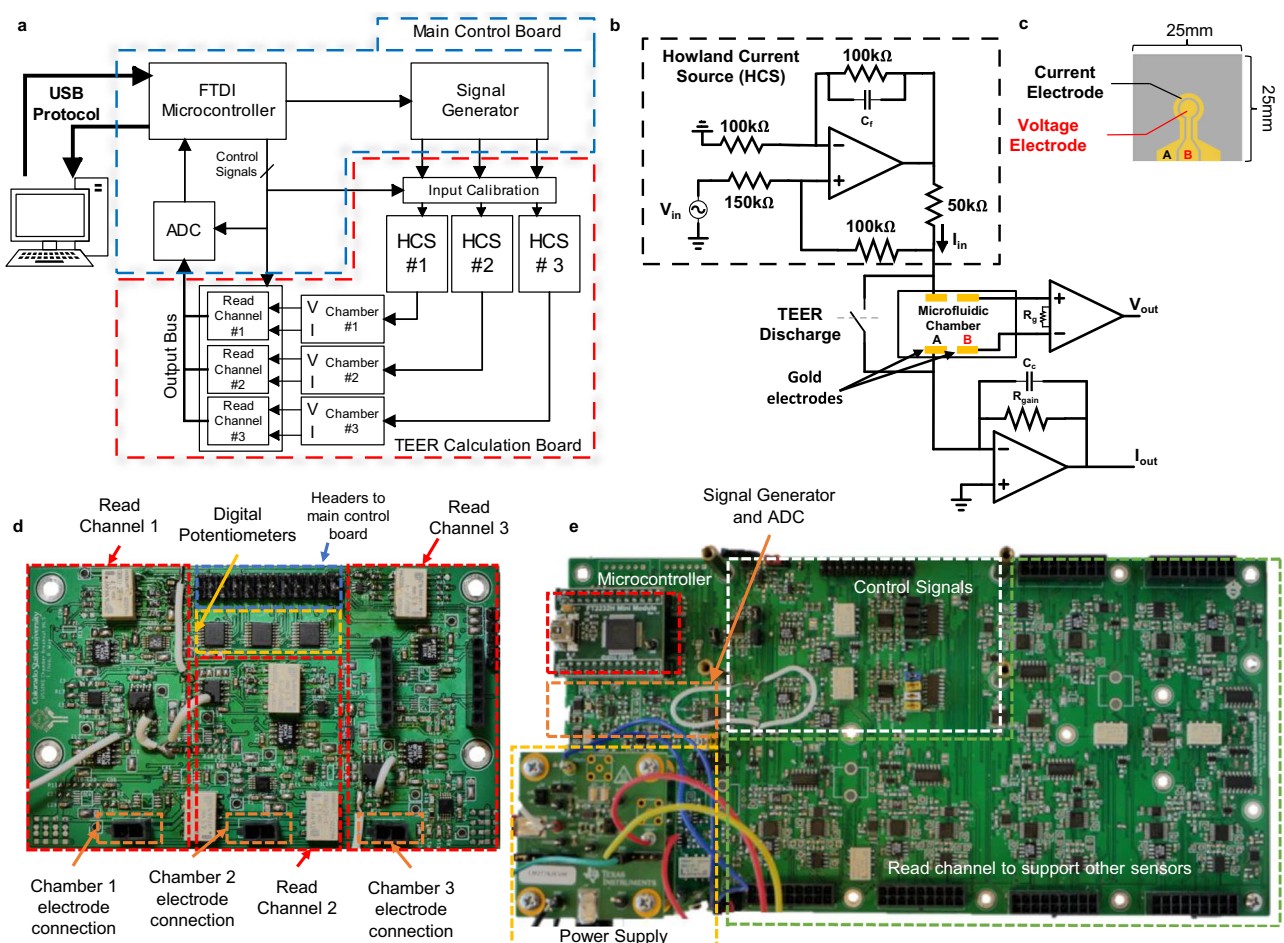

**Fig. 2 | Microphysiological system's supporting electronics. a** Block diagram of signal conditioning and processing flow. **b** Howland current source (HCS) and read channel circuit, responsible for supplying stimulus signal and reading output voltage and current signals directly from the sensor electrodes inside the chamber. **c** The sensor electrodes on glass substrate, outer ring electrode is the current electrode, and the middle circle is the voltage electrode. **d** Transepithelial epithelial electrical resistance (TEER) acquisition board contains all read channels, digital potentiometers, connectors for the electrodes, and connectors to the main control board. **e** Main control board which has the power supply, signal generator and ADC, microcontroller, and control signals.

are used for the three independent chambers in the system. The read-channel (Fig. 2b) consists of a transimpedance amplifier to convert the input current signal to an output voltage, and an instrumentation amplifier to acquire the voltage response from the tissue barrier. A relay is placed in parallel with each microfluidic chamber to discharge built-up charge on the TEER electrodes when necessary. Built-up charge on the electrodes is capable of altering the DC voltage at the input to the microfluidic chamber, and therefore, has a large enough effect to shift the DC voltage towards supply rails, reducing the dynamic range of TEER measurement.

The HCS (Fig. 2b) uses an ultra-low offset voltage operation amplifier and consists of five high precision film resistors and a single feedback capacitor for bandwidth control. High output impedance is achieved by resistor matching. The response TEER voltage from the chamber is read by an instrumentation amplifier (INA) with low input bias current. The INA gain is controlled by the resistor $R_g$. The response TEER current from the chamber is read using a transimpedance amplifier (TIA), with the current gain set by $R_{gain}$. An analog-to-digital converter (ADC) is used to convert the amplified analog outputs from the read-channel to a digital signal that is sent to the host computer via the USB port. Due to different operating voltages of different components along the signal chain to achieve the required output dynamic range, electronic level shifting is required at various points in the system. They are performed by operational amplifiers where the voltage gain is controlled by on board resistors and DC shift is adjusted using a digital potentiometer. The digital potentiometer is calibrated before each

measurement using a binary search algorithm to find the smallest DC offset current. The input stimulus was also designed to have a high and low current setting to maximize the system's output dynamic range.

The supporting electronics are partitioned into two separate PCB boards. The main control board (Fig. 2e) contains the external power supply, USB connectors, microcontroller, ADC, signal generators, level shifters for control signals, and connectors to the TEER acquisition board. The TEER acquisition board (Fig. 2d) contains the HCSs, the read channels for each chamber, the calibration digital potentiometers, and the connections for the card edge connectors. Even though the system allows three chambers to be used at a time, the system architecture was designed to be scalable to allow future expansion to accommodate more chambers and sensors.

### Electrode design and manufacturing
The electrodes are manufactured in gold on a glass substrate. Each electrode chip consists of two gold (Au) electrodes to allow 4-point measurement. The electrode chip (Fig. 2c) was fabricated on a $25 \times 25$ mm glass substrate through an in-house photolithography, deposition, and lift-off process. The mask was designed using AutoCAD software (Autodesk, Inc.) and manufactured by Artnet Pro (San Jose, CA). The full photolithography steps are described previously[29].

### Chamber sterilization
To prevent infection during experiments, all components of the microfluidic chamber (chamber body, glass electrode chip, PDMS layers, PCBs,

tubing, and Luer locks) were put through the first round of sterilization protocol:

1. 20-min bath in 1:10 bleach to water ratio.
2. 10-min soapy water bath inside of ultrasonic cleaner.
3. Next, thoroughly rinse with DI water.
4. 45-min bath in 70% ethanol.
5. Finally, thoroughly rinse with DI water and let air dry.

After the first round of sterilization, the chamber was fully assembled with metal screws and clamps that have been autoclaved (30-min, gravity cycle). After all chambers are assembled, the chambers went through low temp gas sterilization and are kept in a sealed bag before use.

### Animals, tissue collection, and media preparation

In all experiments, male C57BL/6 background mice aged 3–4 months were used. Mice were kept on a 12 h light/dark cycle with access to standard chow and water ad libitum. Animal protocols were approved by the Institutional Animal Care and Use Committee (IACUC) at Colorado State University under United States Department of Agriculture (USDA) guidelines.

Mice were deeply anesthetized with isoflurane and terminated via decapitation to prepare for tissue collection. The intestines were removed and immediately placed in 4 °C 1x Krebs buffer (in mM: 2.5 KCl, 2.5 CaCl$_2$, 126 NaCl, 1.2 MgCl$_2$, 1.2 NaH$_2$PO$_4$). To prevent contractions during dissection, the Krebs buffer contained 1 μl 1 mL$^{-1}$ nicardipine (Sigma Aldrich, St. Louis, MO), an L-type calcium ion channel blocker. Colon was then dissected to remove any remaining mesentery. For experiments in which muscle was removed, a 26 G needle was used to gently tease away the muscle layer on the mesenteric edge of the tissue. Tissue was then cut longitudinally using angled vascular scissors to form flat pieces of tissue around ~5 mm.

Adult Neurobasal media was custom made in house with 2% B27 supplement (ThermoFisher scientific, Waltham, WA), 4 mM glucose, 3% 1 M HEPES buffer (Sigma Aldrich, St. Louis, MO), without phenol red. To help maintain the gut microbiome, luminal media contained 0.4 mg ml$^{-1}$ inulin (soluble fiber) and 0.5 M sodium sulfite (oxygen scavenger) to decrease oxygen levels[17]. The serosal media had ambient levels of oxygen creating an oxygen gradient across the tissue which we previously demonstrated[15] is necessary for preservation of microbiome. After 24 h, luminal media for control tissue was not changed. Treatment group luminal media contained 5.80*10$^{-2}$ U of broad spectrum bacterially sourced collagenase (Worthington Biochemical, Lakewood, NJ) or was treated with hydrochloric acid (HCl) to acidify the pH to 2. After completion of experiments, 0.05 M phosphate buffered saline (PBS) containing 0.5% cetylpyridinium chloride (CPC) was gently pipetted onto the tissue to preserve the mucus layer[12]. The tissue was then gently removed from the device and placed in 4% paraformaldehyde (PFA) containing 0.5% CPC at 4 °C for 24 h. Tissue was stored in PBS at 4 °C until sectioning.

### Tissue sectioning and histochemistry

Detailed methodology can be found in our previous publication[12]. Briefly, 1–3 mm sections of colon were submerged in agarose until polymerization. Tissue was then cut on a vibrating microtome (VT100S; Leica microsystems, Wetzlar, Germany) at a thickness of 50 μm. For lectin and immunohistochemistry, sections were first washed in 1x PBS, then incubated in 0.1 M glycine followed by PBS washes and incubated in 0.5% sodium borohydride followed by PBS washes. Sections were then blocked in PBS with 5% normal goat serum (NGS; Lampire Biological, Pipersville, PA), 1% hydrogen peroxide, and 0.3% Triton X (TX). Next, sections were placed in PBS containing 0.3% TX and 5% NGS with the appropriate lectin or antibody for 2 days. The lectin used was Ulex Europaeus Agglutinin I conjugated to Rhodamine (UEA-1; Vector Labs) at a concentration of 0.125 μg mL$^{-1}$. Primary antibodies used were anti-claudin1 (Invitrogen) 1:200 and anti-peripherin (Sigma-Aldrich) 1:300. After lectin or primary antibody incubation, sections were washed in PBS with 1% NGS. Sections incubated in primary antibody were then incubated with PBS containing 0.02% TX and Alexa Fluor 594 conjugated to secondary antibodies specific to the species of the primary

antibodies at a 1:500 dilution. Finally, sections were washed in PBS, mounted on slides, and cover slipped. Images were taken using a Zeiss LSM800 upright confocal laser scanning microscope and a 20x (W Plan-Apochromat 20X/1.0 DIC Vis-ir ∞ /0.17) objective or an Olympus BH2 brightfield microscope.

### TEER calculation

Processing of TEER signals involves conditioning steps to reduce noise and other artifacts. The conditioned signals were further processed by applying a curve fitting algorithm to obtain the magnitude and phase of the voltage and current response signals. The impedance magnitude ($|Z|$) and phase difference ($\theta_{diff}$) can then be calculated using Eqs. (1) and (2). Where Av$_{current}$ and Av$_{voltage}$ are the current and voltage gain values and $\theta_{current}$(deg) and $\theta_{voltage}$(deg) are the current and voltage respective phase values.

$$|Z| = \frac{V_{peak}}{I_{peak}} * \frac{Av_{current}}{Av_{voltage}} \tag{1}$$

$$\theta_{diff} = \theta_{voltage}(deg) - \theta_{current}(deg) \tag{2}$$

The magnitude and phase values are determined for each frequency to obtain the impedance spectrum of the tissue sample, commonly referred to as electrical impedance spectroscopy (EIS). Due to its versatility of revealing impedance information across a wide range of frequencies, EIS is a widely-used technique to discover the impedance characteristics of tissue/cell-culture samples in Ussing Chambers, organ-on-a-chip devices, and well inserts[13,18,23,27,30–36].

The sinusoidal curve fitting is necessary to further reduce noise and unwanted artifacts in the acquired TEER signal as it is illustrated in Fig. 3a–c. The smoothed signal can provide more accurate magnitude and phase values for the subsequent TEER calculation (Fig. 3c). The curve fitting algorithm also provides drifting correction to the acquired voltage response signal. Drifting of the response voltage signal is caused by offset DC current from the Howland circuit. This offset DC current builds up charge on the serial capacitance associated with electrode's double layer capacitor, resulting in a constant rate increasing (or decreasing) of the DC voltage at the voltage electrodes from the chamber. This effect can be seen in Fig. 3a. The time dependent DC shift of the sinusoidal signal in Fig. 3a needs to be leveled before the sinusoidal curve fitting algorithm can be applied to obtain its magnitude and phase. This is done by subtracting a 1$^{st}$-order polynomial function from the acquired (drifted) voltage signal as illustrated in Fig. 3b.

The TEER value of an epithelial barrier is the resistance of the transcellular and paracellular pathways combined. However, the TEER values obtained from Eqs. (1) and (2) include additional impedance such as the electrode double-layer capacitance and the media bulk resistance[3,10,34]. In order to obtain the actual TEER values associated with the epithelial barrier, baseline TEER measurements were performed for each experiment to capture the medial bulk resistance. The final TEER value of interest was obtained by subtracting the baseline TEER values from the acquired TEER values. It should be noted that the magnitude $|Z|$ used for TEER measurements should be at an appropriate frequency not too low where the impedance of the electrode double-layer capacitance dominates, and also not too high where the epithelial layer is shorted by its parallel capacitance, this can be deduced from the equivalent circuit of the epithelial barrier[10]. From the impedance spectrum of the tissue measured with this device, it was found that this value is close to 5 kHz.

The TEER value can also be calculated by finding the DC response from a square wave. Since the microfluidic chamber system is also capable of producing a square wave stimulus signal, the TEER using the square waveform stimulus was also calculated. This value shows the pure resistance of the tissue barrier. Figure 3d, e shows a set of TEER values obtained using a 5 kHz sinusoidal stimulus vs. a square wave stimulus. It was found that the TEER values obtained using the square waveform are lower (7.2% on average, $n = 10$) than those obtained using the 5 kHz sinusoidal waveform by a constant margin. This is due to the fact that the relatively fast transitions

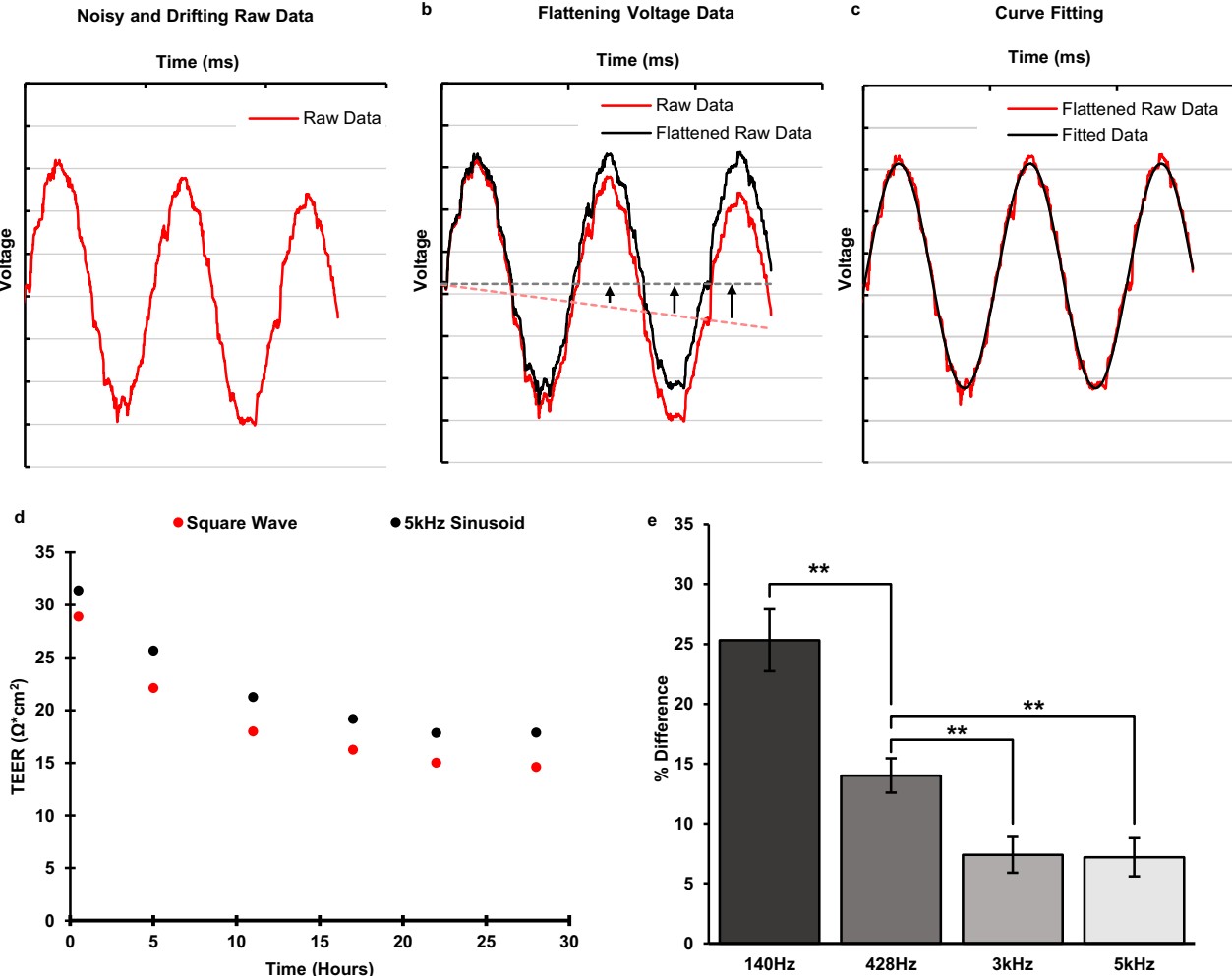

**Fig. 3 | Response signal conditioning and the effect of stimulus signal on TEER calculation. a** Raw response signal with unwanted artifacts. The raw data is noisy and drifting over time because of the offset DC from the Howland current source (HCS). The curve-fitted signal (black) removes the unwanted artifacts from the acquired response signal (red). **b** An example illustrating the drift correction performed by the fitting algorithm. The 1st degree polynomial (dotted red) of the input data is subtracted point by point from the input raw data (solid red), effectively flattening it out. Once the input data is flattened out it can be fit to a sinusoidal curve (black). **c** Curve fitting algorithm applied to flattened raw data (red) to produce noise-free fitted signal (black). **d** Comparison of transepithelial electrical resistance (TEER) measured with square wave and 5 kHz sinusoidal stimulus signals. The square wave stimulus is consistently lower than the sinusoidal value. **e** The average difference of TEER between sinusoidal and square waveforms for different frequencies of the sinusoidal waveforms, $n = 10$, error bars show the standard error, **$P < 0.005$.

in the square waveform stimulus were able to reduce the effect of the double layer capacitance associated with the electrodes on TEER magnitudes compare to that from the 5 kHz sinusoidal stimulus. If the sinusoidal stimulus frequency is decreased, then the effect of the double layer capacitance is more pronounced, making the TEER value increase as the input frequency decreases. Figure 3e confirms this by showing the percent difference between the sinusoidal and square wave increases as the frequency of the sinusoidal stimulus decreases. When the stimulus frequency reaches 3 kHz and above, the difference between sinusoidal and square wave data flattens out, indicating that the stimulus frequency is now high enough to bypass the double layer capacitance.

### Experiment and measurement procedure

After all tissue explants were cut and prepared according to the protocol in section "Animals, Tissue Collection, and Media Preparation", the explants were loaded into the microfluidic chamber, one by one. First, the explants were placed on the bottom half chamber and then gently flattened out using forceps, careful not to touch the luminal side and damage the mucosa. After the tissue was flattened and centered over the holding cavity (Fig. 1c) on the bottom half chamber, the top chamber was slid down the metal screw guides

to secure the tissue in place and create a tight seal. The chamber was then tightened using wing nuts and inserted into the card edge connector on the metal enclosure of the system (Fig. 1g). Next the inlet and outlet tubing were connected. The media outlet tubes fed into empty glass bottles as a way to determine whether even media outlet from each side of the tissue was achieved during experiments. To remove any air bubbles in the chamber the media was purged into the chamber at an increased rate (25,000 μL h$^{-1}$) for 45 seconds at the start of each experiment. After the initial purge, the media flow rate was reduced to 250 μL h$^{-1}$ throughout the experiment. The chambers, media, and the system enclosure are all kept in an incubator set to 37 °C.

To evaluate the live tissue viability, tissues were kept inside of the device for up to 72 h with fresh control media being pumped through continuously. Tissue viability was assessed by examining the tissue with histochemical methods and searching for characteristics of healthy tissue. During TEER experiments tissues were kept under control conditions for the first 24 h. After this calibration period, tissues were exposed to different media compositions, either collagenase or acidic media (discussed in section "Animals, Tissue Collection, and Media Preparation") for the next 24 h. Over the course of the full experiment TEER measurements are performed every 2 h,

creating a timeline of the full 48 h experiment. For each TEER measurement the input AC current magnitude was set at 85 μA and the frequency was swept from 12 Hz to 5 kHz at 20 different frequency points. At each measurement point, the TEER was measured using the sinusoidal waveform stimulus as well as the square waveform stimulus. After the TEER experiments the tissue follows the same protocol as the tissue viability experiments, this protocol is outlined in section "Tissue Sectioning and Histochemistry".

For experiments comparing the reduction in TEER after being exposed to collagenase or acidic media the reduction is measured from after the calibration period (~24 h) to the end of the experiment (48 h). Only TEER values from the 24 h – 48 h mark are considered for the experiment because the first 24 h is considered a good calibration period for the tissues. We found that TEER values found during the first 24 h provide sporadic results as the tissue reaches equilibrium in the device environment around 24 h as the TEER signal settles. The TEER experiments were not considered past 48 h because of the degradation witnessed in the low pH exposed tissue and the same method relying on sectioning and imaging tissues after the experiments was used for viability for consistency.

## Reporting summary

Further information on research design is available in the Nature Portfolio Reporting Summary linked to this article.

## Results and discussion
### Results of system electrical and noise performance

The frequency response of each circuit component along the signal path is shown in Fig. 4a–d. The component with the lowest bandwidth of 47.5 kHz is the TIA (Fig. 4d). The bandwidth of the TIA was set by a compensation capacitor to be roughly ten times higher than the highest frequency of input sinusoidal stimulus (5 kHz). This did not attenuate any important read channel signals while also filtering out as much high frequency noise as possible. It should be noted that the TIA tends to have high input inferred noise due to the high thermal noise of its gain-setting resistors. It sits relatively late in the analog signal chain and its low bandwidth can filter out the output noise from other components (input level shifter and HCS) before it in the signal chain. This sets the full systems bandwidth at 47.5 kHz as intended.

The stability of each read channel is examined by its step response to obtain sufficiently damped responses (Fig. 4f). The noise power spectral density (PSD) of the read channel was measured and shown in Fig. 4e. The results show the total noise power to be 0.126 μV2, well below the minimum output signal power of 82.1 μV2 of the system, resulting in a signal-to-noise ratio (SNR) of 28.14 dB. Other system performance parameters, such as power consumption, TEER measurement error, and the acceptable TEER range with error less than 5%, were also measured and calculated. Table 1 summarizes the system level electrical performance of the microphysiological system.

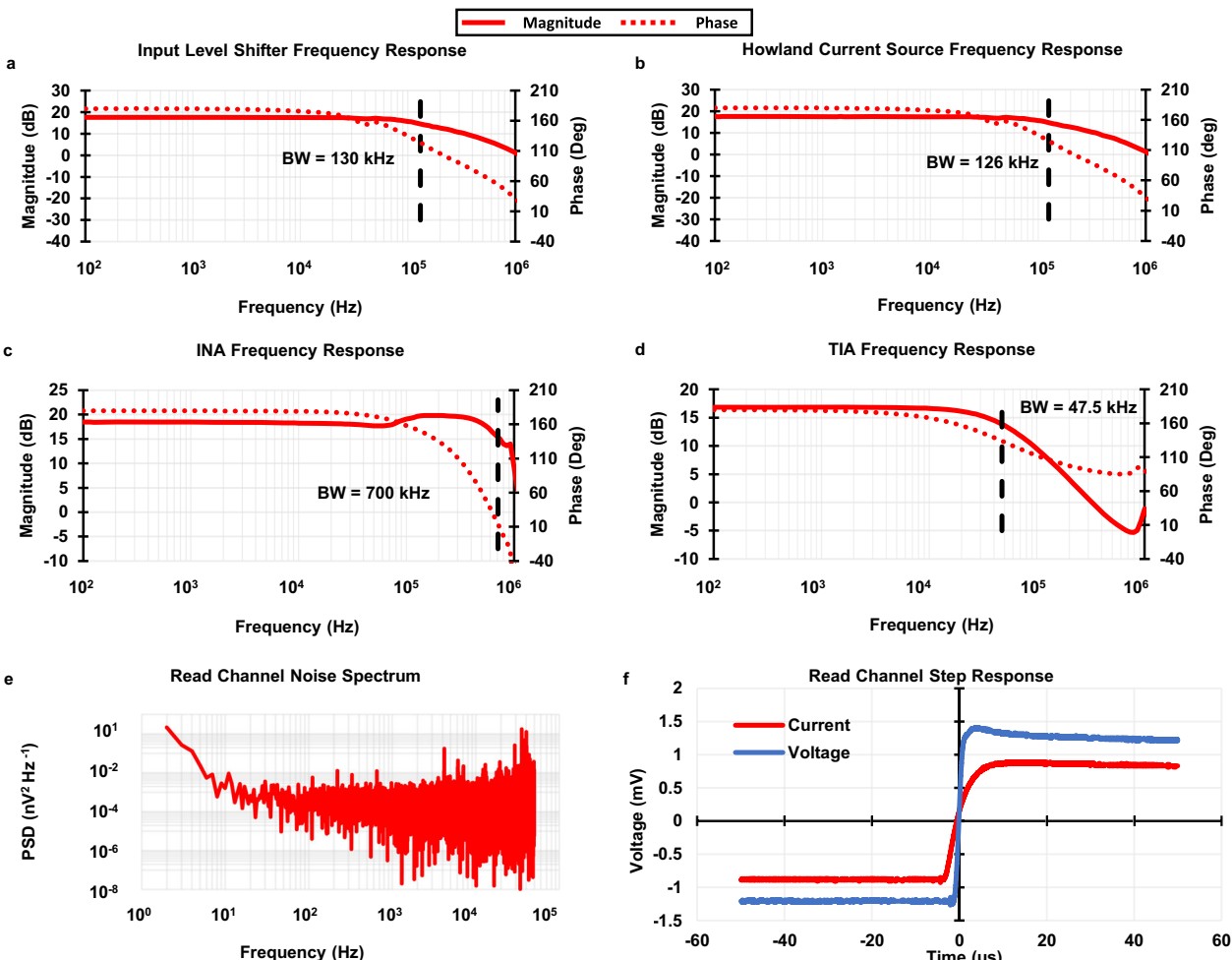

**Fig. 4 | System electrical performance.** The frequency response for each component in the signal path (**a–d**) was used to find the systems bandwidth. **a** Input level shifter frequency response. **b** Howland current source (HCS) frequency response. **c** The Instrumentation amplifier (INA) frequency response. **d** Transimpedance amplifier (TIA) frequency response. The limiting component is the TIA (**d**), this component sets the systems bandwidth at 47.5 kHz. **e** The read channel noise power spectral density (PSD) was found to classify the noise specification of the system. **f** The step response of the voltage and current stages in the read channel. The lack of ringing in both step responses confirm the stability of the read channel.

## Table 1 | Full system specifications

| Specification | Value | Unit |
|---|---|---|
| Impedance Calculation | | |
| Frequency Range | 10–5k | Hz |
| Impedance Range (Error <5%) | 150–6.5k | Ω |
| Sampling | | |
| ADC Sampling Rate | 806.4k | samples/s |
| Resolution | 12 | bits |
| Power Consumption | | |
| $V_{dd}$ | ±5 | V |
| Full system | 4.203 | W |
| TEER Circuit Add on | 1.17 | W |
| Signal Processing | | |
| Bandwidth | 47.5 | kHz |
| Howland Offset Current (DC) | 1.87 | µA |
| Voltage Gain | 9.8214 | gain |
| Current Gain | 27,000 | gain |
| Noise | | |
| SNR | 28.14 | dB |
| Total Noise Power | 0.1261 | µV2 |
| Avg. Spectral Density | 623.824 | $nV/\sqrt{Hz}$ |
| Spot Noise @ 100 Hz | 1685.79 | $nV/\sqrt{Hz}$ |
| Spot Noise @ 1 kHz | 630.688 | $nV/\sqrt{Hz}$ |
| Spot Noise @ 10 kHz | 762.691 | $nV/\sqrt{Hz}$ |

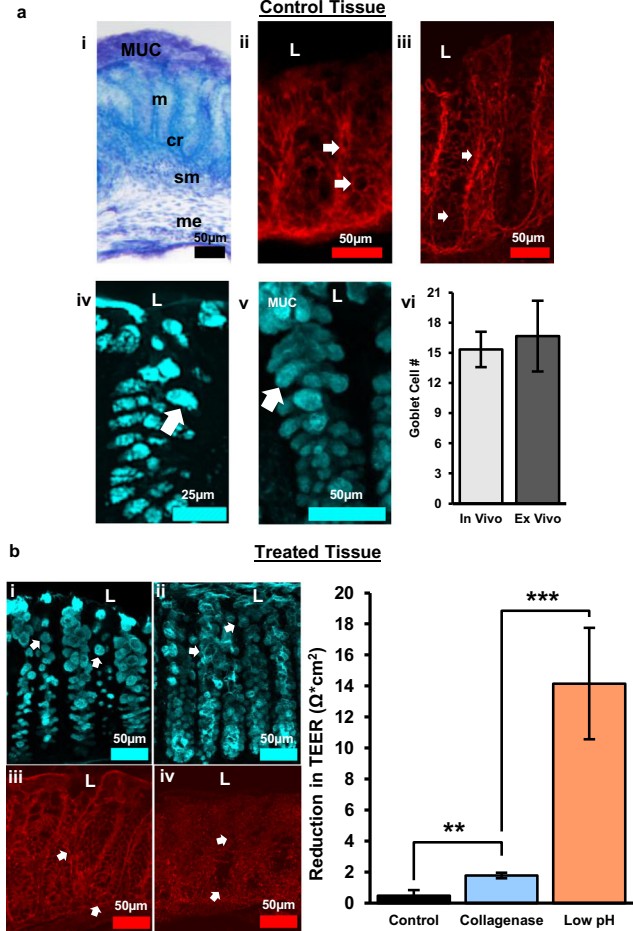

**Fig. 5 | Tissue health was maintained over 72 h in the device and monitored after media treatment. a** Control tissue after 72 h experiment. (i) Tol blue staining showing maintenance of colon morphology, MUC = mucus layer, m = mucosa, cr = crypt, sm = submucosa, me = muscularis externa. (ii) Claudin-1 immunoreactivity in vivo. (iii) Claudin-1 immunoreactivity in 72 h ex vivo shows maintenance of tight junctions between epithelial cells and crypts. (iv) UEA-1⁺ material in vivo. (v) UEA-1⁺ material ex vivo confirming maintenance of epithelial cells and mucus layer. (vi) Quantification of goblet cell number per apical crypt in in vivo vs. ex vivo tissue, $n = 3$, error bars show the standard error. **b** Collagenase treated, and acidic luminal media resulted in alterations in goblet cell morphology and tight junction expression indicative of increased barrier permeability. (i) Goblet cells labeled with UEA-1 become circular after collagenase treatment. (ii) Acidic media resulted in loss of goblet cell shape and sloughing off of cells near the lumen. (iii) Alterations in tight junction protein expression (claudin-1) following collagenase treatment. (iv) Claudin-1 expression decreased considerably with exposure to acidic media indicative of substantial barrier disruption. (v) The bar graph shows a distinct reduction in transepithelial electrical resistance (TEER) after exposure to different media composition. The difference in TEER was measured from 24 to 48 h mark after the tissue was enclosed in the device, with the media change occurring at 24 h. The three media compositions consist of a control media, collagenase treated media, and low pH media (more details about media composition in "*Animals, Tissue Collection, and Media Preparation*"). L = lumen, TEER values are normalized to the membrane surface area of the chamber, 0.0314 cm², Control: $n = 4$, Collagenase: $n = 10$, Low pH: $n = 3$, error bars show standard error, **$p < 0.005$; ***$p < 0.0001$.

## Tissue viability

Tissue health was maintained in the device with barrier integrity over 72 h. Colon explants maintained proper arrangement of mucosal, submucosal, muscular layers, and patterned crypts and the inner mucosal layer remained intact confirming maintenance of the mucosal barrier (Fig. 5a(i)). To protect the body from potential pathogens, healthy intestinal tissue must maintain sophisticated epithelial and mucosal barriers. Specialized epithelial cells, known as goblet cells, are crucial to barrier maintenance as they are responsible for producing and secreting mucin. Goblet cells were characterized due to their essential roles in maintenance of the barrier. Goblet cell mucopolysaccharides were identified by binding *Ulex europaeus* agglutinin I (UEA-1) conjugated to rhodamine. After 72 h in the microphysiological system, goblet cells retained their distinct shape (Fig. 5a(v), arrows) comparable to tissue taken directly after sacrifice that never went into the device (in vivo tissue) (Fig. 5a(iv), arrows). A hallmark indicator of intestinal tissue damage in several disorders is a decrease in number of goblet cells[37]. Goblet cells arise from progenitor cells at the base of colonic crypts. As they migrate apically, they mature and acquire the ability to synthesize and secrete mucopolysaccharides[38]. Since mature goblet cells are located in the upper third of crypts[39], they are potentially more susceptible to morphological changes. Therefore, the number of goblet cells in the upper third of crypts (apical crypt) were analyzed to further confirm tissue health. There were no statistically significant differences in the number of apical goblet cells between in vivo and ex vivo tissue (Fig. 5a(vi)) indicating that 72 h ex vivo tissue was likely comparable in health to in vivo tissue. To further assess barrier integrity, tight junctions were examined. Tight junctions adhere epithelial cells together forming a physical barrier between cells to prevent unwanted passage of ions and molecules between epithelial cells. Claudins are a specific type of tight junction protein that help form the backbone of tight junctions. Claudin-1 is widely expressed in the intestinal epithelium and has essential roles in tight junction integrity. After 72 h in the device, clear claudin-1 immunoreactivity remained around epithelial cells

(Fig. 5a(iii)) similar to in vivo tissue (Fig. 5a(ii)) further indicating maintenance of tissue health and barrier integrity.

## Using TEER to measure changes in barrier permeability

Changes in TEER were correlated with physiological signs of barrier impairment, such as alterations to epithelial cells, the mucus layer, and tight junction proteins. To induce a disruption to barrier permeability, the

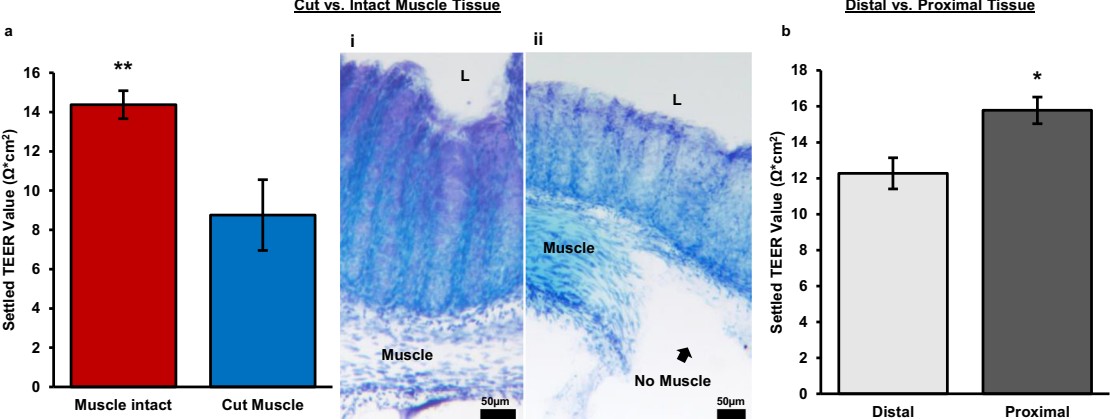

**Fig. 6 | Physical differences in tissue explant. a** This bar chart shows the difference in settled transepithelial electrical resistance (TEER) value of tissues explants with the muscle intact vs. with the muscle removed. The settled TEER value is taken 24 h after the tissue is enclosed in the chamber. (i) Tol blue staining of tissue with intact muscle. (ii) Tol blue staining tissue with muscle removed, L = lumen. **b** Settled TEER values for different regions of mouse colon tissue. Proximal tissue was defined as the three pieces of tissue closest to the cecum and distal tissue was defined as the two pieces farthest from the cecum and closest to the rectum. Each tissue piece was approximately 5 mm in length. The settled TEER value was taken approximately 24 h after the tissue had been enclosed in the device. Muscle intact: $n = 15$, Cut muscle: $n = 7$, Distal: $n = 6$, Proximal: $n = 9$, error bars show standard error, **$P < 0.005$, *$P < 0.01$.

luminal side of colon tissue was treated with collagenase or acidic media. Bacterial collagenases are enzymes secreted by endogenous bacteria in the intestines that degrade collagen. Increased collagenase can break down tight junctions between epithelial cells, as well as, break down the extracellular matrix of epithelial cells[40]. This leads to increased intestinal permeability, and provides a model for the development of leaky gut syndrome[41]. We have previously shown that bacterial collagenase in luminal media can be used as a model to create leaky gut by disrupting epithelial cell (goblet cell) morphology and decreasing tight junction (claudin-1) expression[12]. Increased barrier permeability was shown by an increased reduction in TEER with collagenase treatment over time (Fig. 5b(v)). To confirm that reductions in TEER correlated with physiological characteristics of increased intestinal permeability, goblet cells, and claudin-1 were examined. Following collagenase treatment, goblet cells became more circular in shape (Fig. 5b(i)) and claudin-1 immunoreactivity was moderately decreased (Fig. 5b(iii)) indicating barrier impairment.

To assess whether changes in TEER matched changes in physiological changes, acidic media (pH 2) was added to the luminal side of tissue to induce substantial damage to the intestinal barrier. Cells need to maintain a pH of 7.4 to function properly. Lowering the pH to 2.0 leads to increased epithelial cell death and alterations in cellular processes creating drastic increases in permeability. This was confirmed with goblet cells losing distinct shape and sloughing off near the lumen (Fig. 5b(ii)). Claudin-1 immunoreactivity dramatically decreased indicating substantial loss of tight junctions (Fig. 5b(iv)). These dramatic changes in epithelial cell and claudin-1 morphology correlate with the significant reduction of TEER following acidic pH treatment (Fig. 5b(v)).

### Differences in tissue explant detected by TEER
**Cut muscle vs. muscle intact muscle.** To determine whether distinct tissue components contributed differentially to TEER, the muscle layer was dissected away. Thereby removing the muscularis externa, a major subepithelial structure of the colon. TEER was measured after 24 h inside the chamber, allowing sufficient time for the tissue to equilibrate to its new environment. Removal of the muscle layer decreased TEER by about 39% (Fig. 6a). This result is consistent with previous reports that have performed experiments to study the contribution of sub epithelial resistance. The values reported have ranged from 15% to 80% of the total epithelial resistance is concentrated in the sub epithelium, depending on the location in the intestine as well as the animal[23,34,42–44]. Research using rat jejunum has shown a much larger contribution to total resistance

done by the sub epithelium (78–80%)[23,42]. Whereas measurements on the ileum, colon, and rectum in both rats and mice have showed much lower contributions (15–45%)[34,43,44]. This is consistent with the results found here using mouse colon.

Confirmation of total muscle dissection was done by Toluidine blue staining, as seen in Fig. 6a(ii) when compared to tissue with the muscle intact (Fig. 6a(i)). Demonstrating the TEER calculated with the muscle dissected is an accurate representation of the epithelial resistance alone and has little to no contribution from subepithelial resistances. This demonstration is lacking from all previous research studying the contribution of subepithelial resistance by subepithelial stripping or dissection[42–44]. These images alongside the TEER measurements provide new evidence for the contribution of subepithelial resistance to total epithelial resistance.

**Proximal vs. distal colon.** The position within the colon is another possible factor contributing to differences among tissue explant slices. Previous research on mouse colon tissue has shown that the proximal tissue has a higher baseline TEER value compared to mid or distal slices[21]. After compiling data from our own experiments, we found that our TEER results are consistent with previously reported data on tissues located in different sections of the colon. We observed about a 19% decrease in TEER in the distal colon compared to the proximal colon (Fig. 6b).

### Highlights of system performance
To showcase the distinctive qualities of our system, we compared it with existing devices for studying epithelial transport based on several key attributes, such as sample type, tissue viability duration, TEER measurement capability, experiment throughput, and more. The comparison is shown in Table 2. All of these attributes are important for evaluating different epithelial transport systems, and highlighting the differences between Transwell, OoC, and ex vivo devices.

The top four rows of the table show existing in vitro devices (Transwell and OoC). These systems are intended for studying cultured cell monolayers. In addition to providing TEER measurements, they also provide high throughput of up to 96 wells. Unlike the Transwell system, most of the OoC systems provide microfluidic support to maintain cell viability. One of their biggest weaknesses is their lack of realistic biological model to better represent rich biological processes in live tissues.

The remaining rows in Table 2 are existing ex vivo systems. Our proposed system is in the bottom row. Some of the existing ex vivo systems are capable of TEER measurements, but they can only maintain tissue

**Table 2 | Comparison of epithelial barrier investigation devices**

| | Biological Sample | | Electrical Permeability | | | | Chamber/System Design | |
|---|---|---|---|---|---|---|---|---|
| | Sample Type | Demonstrated Tissue Viability | TEER capable? | Electrode Type | Stimulus Signal | Measurement Electronics | Microfluidics Support | Throughput |
| Transwell[2,16] | Cell monolayer | - | Yes | Ag/AgCl "stick" electrodes | DC | Commercial Benchtop | No | 96 |
| Liang et al.[13] | Cell monolayer (canine kidney) | - | Yes | Integrated glass chip | Up to 10 MHz | Commercial Benchtop | Yes | 1 |
| Helm et al.[18] | Cell monolayer (Caco-2) | - | Yes | Polycarbonate substrate electrode chips | Up to 100 kHz | Commercial Benchtop | Yes | 1 |
| Fernandes et al.[35] | Cell monolayer (GI tract and airway) | - | Yes | Integrated glass chip | Up to 100 kHz | Custom-built | One side only | 8 |
| Navicyte[21] | Mouse and human intestinal tissue | <3 h | Yes | Ag/AgCl "stick" electrodes | DC | Commercial Benchtop | No | 6 |
| Clarke et al.[20] | Mouse colon tissue | 3 h | Yes | Ag/AgCl electrodes connected by salt bridge | DC | Commercial Benchtop | No | 1 |
| Calvo et al.[27] | Frog epithelial tissue | - | Yes | Integrated "stick" electrodes | Up to 100 kHz | Custom-built | No | 1 |
| Dawson et al.[25] | Human intestine tissue | 72 h | No | - | - | - | Yes | 1 |
| Poenar et al.[22] | Porcine esophageal tissue | 48 | Yes | Integrated "stick" electrodes | DC | Commercial Benchtop | Yes | 1 |
| Cherwin et al.[12] & Richardson et al.[15] | Mouse colon tissue | 72 h | No | - | - | - | Yes | 1 |
| Amirabadi et al.[26] | Porcine & human colon tissue | 24 h | No | Optical Fiber Sensor | - | - | Yes | 1 |
| This Work | Mouse colon tissue | 72 h | Yes | Integrated glass chip | Up to 5 kHz | Custom-built | Yes | 3 |

viability for a short period of time (<3 h). The ex vivo devices that are capable of maintaining tissue viability for longer period of time (up to 72 h) are not capable of TEER measurements or any other real-time tissue integrity feedback measurements. Furthermore, the existing ex vivo systems can only support one tissue measurement at a time, making experiments requiring replicates more difficult and time consuming. In contrast, our proposed system is designed to perform real-time TEER measurements on multiple tissue explants at the same time, with the capability of maintaining tissue viability for up to 72 h. These advances in capabilities will make studies targeting a specific organ, requiring extended experiment time period of interrogation, and replicates from the same animal, easier and more feasible.

## Conclusion

This paper presents a highly integrated microphysiological system for studying live tissue barrier permeability of mouse colon. The unique design of the microfluidic chamber is capable of securing an explant of mouse colon tissue between two independent media pathways creating a micro physiological environment inside the chamber comparable to the environment in vivo. The use of proper media provides nutrients, support gut microbiome, and create important oxygen gradients across the tissue to keep tissue viability for an extended period of time. After 72 h in the chamber, the tissue explants displayed an inner mucus layer, robust goblet cells, and evident tight junction function along the length of the epithelial layer. These characteristics all serve as strong indicators of sustained barrier integrity. This preservation of tissue viability addresses an important drawback in existing live tissue barrier permeability devices.

Integrated electrode chips allow the microfluidic chamber to successfully characterize barrier permeability using TEER measurements in real time. The plug-and-play nature of the system design simplifies the experiment setup and allows for all chambers to be re-usable and universal. Unlike most existing systems where bulky and expensive benchtop equipment is needed to perform experiments, the integrated support electronics made the overall system small enough to fit into an incubator. Furthermore, architectural scalability allows multiple chambers to be connected to the system enabling multiple controlled experiments using samples from the same donor. The use of the system is further enhanced by a custom-built GUI which was developed to allow each experiment to be customizable and ran from any host computer.

In conclusion, this microphysiological system has the potential to open new avenues to investigate barrier health of live tissues. Real time permeability measurements are crucial to developing more accurate ex vivo tissue models for studying the health and chemical responses of barrier tissues.

## Data availability

Data underlying the results presented in this paper are not publicly available but may be obtained from the authors upon reasonable request.

## Code availability

All custom code used in the research presented in this paper are available upon request from the corresponding author. All curve fitting algorithms used in the results of this paper are based off the publicly available Python package "scipy.optimize.curve_fit". Link to the package's user manual provided on SciPy's website: https://docs.scipy.org/doc/scipy/reference/generated/scipy.optimize.curve_fit.html.

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

## Acknowledgements
The authors would like to thank Amanda Cherwin, Alexis Ehrlich, and Connie King for their contributions in tissue sectioning/imagery and dissection, as well as useful discussions on microfluidic device design. We thank Dr. Luke Schwerdtfeger for providing helpful comments on the manuscript. Results presented in this paper are based upon collaborative work partially supported by National Science Foundation (USA) Grants no. 0841259 and 1450032. Any opinions, findings, conclusions, or recommendations expressed in this paper are those of the author(s) and do not necessarily reflect the views of the National Science Foundation (USA). SAT was supported in part by ORWH-NIMH U54-MH118919 (SAT and Jill Goldstein, Multi-PIs).

## Author contributions
Ryan Way: investigation, software, validation, methodology, data curation, writing – original draft. Hayley Templeton: investigation, validation, methodology, data curation, writing – original draft. Daniel Ball: investigation. Ming-Hao Cheng: investigation. Stuart A. Tobet: supervision, methodology, writing – reviewing & editing, funding. Thomas Chen: supervision, methodology, writing – reviewing & editing, funding.

## Competing interests
The authors declare no competing interests.
