## [Peer Review File · Communications Engineering]

This file contains all reviewer reports in order by version, followed by all author rebuttals in order by version.Reviewers' comments:

Reviewer #1 (Remarks to the Author):

The paper describes a new device that is capable of measuring the TEER of intestinal mouse tissue slices. The device has a couple of advantages over previous devices, most importantly, it is capable of keeping the tissue slices alive for up to 72 hours, which has not been previously achieved in devices that are capable of measuring TEER. However, this is important because the devices make it possible to measure not only short term, but also somewhat longer-term influences of drugs on the barrier function of the intestine (and other factors that influence the intestinal barrier function).

The described experiments are well-conceived and executed, and the paper is well-written.

I only have a few minor comments:

- 1) Figure 3a, b, and c are missing the time unit.
- 2) Figure 3d is using "Hours" as time unit. I would suggest changing it to the SI version: "h".
- 3) For the reduction shown in Fig. 5b, I'm assuming it is a x-fold reduction?

Reviewer #2 (Remarks to the Author):

This manuscript by Way et al describes a microphysiological system that enables real-time recording of gut epithelial barrier permeability for in vitro measurement of transepithelial electrical resistance (TEER). The authors introduce engineering approaches to i) create a microfluidic chamber for maintaining viability and functions of tissue explants, ii) integrate sensor electrodes and supporting electronics for acquiring TEER measurements, and iii) increase scalability of the system to allow multiple chambers running simultaneously. The proof-of-principle for this strategy is demonstrated by placing mice gut tissue explants into the microfluidic chamber to measure TEER.

Overall this work provides conceptual and technological advances with the potential for broad applications in measuring barrier function and permeability in tissue explants. However, the authors should conduct more thorough and systematic control studies to show whether and how

their strategy is more beneficial over existing methods. First of all, major concerns arise from poor justification of this study. In the introduction, the authors describe recent efforts to measure TEER of the epithelial barrier using ex vivo systems and then suggest a “gap” between recently developed organ-on-a-chip in vitro models and their limited capability as an in vitro cell culture model to model epithelial barrier as the primary driver of the current study. Considering that the vast majority of tissue-on-chip models discussed here are intended for in vitro screening and bioassay applications, this statement is misleading and raises critical questions regarding the significance and rationale of this study. Key questions that remain to be addressed include:

- Why is it even necessary to consider ex vivo TEER measurement of tissue chips?
- Alternatively, what types of studies have been done to integrate TEER sensors or permeability assays into organ-on-a-chip in vitro models? What are the specific examples and how significant are they?
- Can the technical challenges associated with integrating ex vivo measurements of epithelial permeability into organ-on-a-chip in vitro models be regarded as an important barrier to progress in biomedical research and drug discovery (as suggested by the authors)?

More detailed comments:

1. The authors fail to provide a strong motivation for the use of an ex vivo TEER measurement system. What additional information do they wish to gain, and is this information specific to this intestine explant model? A major revision of the introduction is suggested with a stronger emphasis on the benefits of epithelial permeability and TEER measurements of tissue explants, more background on intestinal biology, and the motivation to translate this model to drug discovery and development research.
2. This manuscript would benefit from more logical flow and transitions between sections and within each section in the introduction. An example of disjointed points is the jump from TEER to organ-on-a-chip model systems on page 2. More information is needed to make this transition clear.
3. Tissue-on-a-chip and organ-on-a-chip models are being developed to provide new opportunities to recapitulate in vivo environments and to enable high-content analysis without the use of animal models or humans. This is the benefit of many of these systems. On page 2, the authors suggest that the lack of in vivo-like epithelial barrier tissue complexity is a limitation to these models. Again, can the authors provide specific information as to why this is and what information would be gained with ex vivo tissue explant models?

4. The techniques described here may not translate well to a scalable high-throughput system. Also, tissue explants can only survive for 72 hours in the current system. Authors should provide more experimental data to clarify the potential use of their system as a high-throughput live tissue assessment platform.

5. The bulk of the results section focuses on TEER system set up. Discussion of the biological results (especially those from intestinal tissues) is often phenomenological and contains technical details that would be more appropriate for supporting information. More space should be dedicated to discussing the physiological relevance and significance of their approaches and major findings from the experimental studies.

Reviewer #3 (Remarks to the Author):

This manuscript describes the development of a microfluidic platform integrated with electrodes that enable TEER measurements for various types of ex vivo animal epithelial tissues. Compared with existing methods, this approach uniquely supports the long-term culture of ex vivo animal tissues using microfluidics, while the incorporated electrodes allow for real-time monitoring of their TEER values. Overall, the study was well-organized and established as the authors presented sufficient methodological details and good-quality results. However, to further enhance the robustness of the study, I would like to raise some questions and offer suggestions, in particular for long-term ex vivo culture capability of this approach and the material biocompatibility.

First, in the Introduction section, the authors claim that their approach allows for the long-term culture of ex vivo tissues compared with existing methods. However, the authors did not provide sufficient data or rationale to explain why this happens with their approach. While the application of microfluidics may be an enabling factor, additional supporting data are needed. For example, could you provide and compare tissue viability tests for microfluidic and static cultures in the proposed device?

Second, regarding the section of tissue viability, I have a couple of suggestions. This section is important as it validates the efficacy of the approach, especially for the long-term culture of ex vivo tissues. However, the authors have provided only qualitative analyses of animal colon epithelium

three days post-culture, without any comparison to control groups. Please consider comparing the post-three-day tissues with the initial ex vivo epithelial tissues that are uncultured in the devices.

Additionally, alongside studies on tight junction degradation and goblet cell functions, could you also provide data on actual epithelial cell viability? This would directly demonstrate the ability of this approach to maintain the viability of ex vivo epithelial cells.

Regarding Figure 5v, could you provide the temporal changes in TEER at 0, 24, 48, and 72 hours for the various media compositions, instead of only showing changes between 24 and 48 hours? This additional data would validate the efficacy and reliability of this approach by demonstrating whether all sample tissues start with similar TEER values and confirming that this approach can function effectively for 72 hours.

I am also concerned about the biocompatibility of the Anycubics UV resin used to construct the microfluidic chamber. While the authors attempted to mitigate any harmful effects by ensuring the resin is fully cured, I believe that the biocompatibility of the resin still remains uncertain. To address this issue, I suggest conducting a straightforward experiment comparing cell viability between tissues (or any type cells) cultured in wells made from the proposed resin and those cultured in commercially available well plates.

In the section discussing the proximal vs. distal colon, the difference in TEER between these colon segments is attributed to their in vivo pH differences. However, it appears that the same pH level of the tissue culture media and their microfluidic flowing were used for both ex vivo tissues in the experimental setup suggesting the pH conditions for both tissue sections might be identical. Therefore, further discussion on this aspect is critical to assess the reliability of the proposed approach.

If these concerns are addressed in the revised manuscript, it would enhance its robustness and scientific credibility.

Reviewers' comments:

Reviewer #2 (Remarks to the Author):

Comments on Highlights of System Performance:

The authors highlight the current system as a high-throughput system due to its capabilities of being a microphysiological system. However, they do not demonstrate this capability in the manuscript. A 3X throughput doesn't sound high throughput in the context of the microfluidic or organ-on-a-chip research field. I suggest that this section should focus on the system's ability to measure TEER values *ex vivo* and discuss why this has advantages over previous TEER measurement systems.

Reviewer #3 (Remarks to the Author):

The authors' responses and the resulting manuscript revision reasonably address the criticisms I raised. Therefore, I recommend the current manuscript for publication in COMMSSENG.

Communications Engineering, Nature Publishing

Response to Reviewers:

Reviewer #1:

1) Figure 3a, b, and c are missing the time unit.

Time unit has been added to the plots.

2) Figure 3d is using “Hours” as time unit. I would suggest changing it to the SI version: “h”.

This change has been made.

3) For the reduction shown in Fig. 5b, I’m assuming it is a x-fold reduction?

The bar graph seen in Fig. 5b shows the linear decrease in TEER from 24h to 48h. The x-axis label has been changed to say ‘decrease’ instead of ‘reduction’ to make this clearer.

Reviewer #2:

- Why is it even necessary to consider ex vivo TEER measurement of tissue chips?

It is necessary to consider ex vivo TEER measurements to evaluate the integrity and functionality of barriers in real time. Without the application of TEER, tissue would need to be taken and removed from the experiment at multiple time points to assess barrier integrity. This would result in the use of more animals to obtain enough tissue to perform long time point experiments (i.e. 72 h) and would significantly increase labor and time to achieve results.

- Alternatively, what types of studies have been done to integrate TEER sensors or permeability assays into organ-on-a-chip in vitro models? What are the specific examples and how significant are they?

There are many examples of TEER sensors being integrated into organ-on-a-chip devices and being used to study different types of in vitro models. Here is a list of a few examples we have cited in our manuscript:

1. Helm et al. created an OoC device that has integrated electrodes and microfluidic channels, they performed all experiments on Caco-2 epithelial cells. They focused on the importance of impedance spectroscopy in cell barrier characterization.
2. Walter et al. also created a device with integrated electrodes and channels for media flow. They were able to successfully co-culture 2-3 types of cells at a time. Using this device, they modeled the intestinal barrier, lung epithelial cell lines, and the blood-brain-barrier depending on the main cell line chosen.
3. Liang et al. cultured Madin-Darby canine kidney cells inside of their OoC device and used real time TEER measurement to study the equivalent electrical circuit of the epithelial barrier. They provide strong evidence for how TEER measurements are accurate representations of the epithelial barrier tightness.

- Can the technical challenges associated with integrating ex vivo measurements of epithelial permeability into organ-on-a-chip in vitro models be regarded as an important barrier to progress in biomedical research and drug discovery (as suggested by the authors)?

Incorporating ex vivo tissue in TEER measurements is important because current models used to measure TEER lack the cellular diversity and complexity seen in vivo which could lead to inaccurate representations of barrier integrity. In the intestine, many factors such as muscle layer, mucus, and the microbiome contribute to TEER. It is important to have a model that most accurately represents in vivo conditions to increase effectiveness and accuracy in the development of drugs and treatments of disease. If key components of a barrier system are missing in drug development, the drug may not perform as well in vivo.

1. The authors fail to provide a strong motivation for the use of an ex vivo TEER measurement system. What additional information do they wish to gain, and is this information specific to this intestine explant model? A major revision of the introduction is suggested with a stronger emphasis on the benefits of epithelial permeability and TEER measurements of tissue explants, more background on intestinal biology, and the motivation to translate this model to drug discovery and development research.

The authors appreciate this feedback. Revisions to the introduction section have been made to highlight the motivations for ex vivo TEER measurement. We discussed that TEER measurements are a widely accepted quantitative technique for measuring barrier integrity and tight junction dynamics for in-vitro epithelial barrier models. Their popularity stems from providing real-time impedance spectroscopy and little damage to the cellular structure of in-vitro barriers. However, in-vitro models lack intricate cell-to-cell and cell-to-matrix interactions found in 3D ex-vivo explant tissue models. This is one of the main limitations of using in-vitro models for biological relevancy compared to ex-vivo explant tissue models. Our goal of this system is to demonstrate the use of TEER for studying ex-vivo tissue explants to allow interrogations of ex-vivo explant tissues with higher biological relevancy. With all the complex cellular interactions being maintained in an ex vivo tissue, our system is better equipped to investigate essential biological questions and hypotheses for drug discovery and development research when compared to in-vitro models.

2. This manuscript would benefit from more logical flow and transitions between sections and within each section in the introduction. An example of disjointed points is the jump from TEER to organ-on-a-chip model systems on page 2. More information is needed to make this transition clear.

The authors agree with this comment. Revisions were made to the introduction to create a better flow to explain the existing devices motivations and their shortcomings in a more logical order. Our goal is to explain how the design of our proposed system was partially inspired by previous devices.

3. Tissue-on-a-chip and organ-on-a-chip models are being developed to provide new opportunities to recapitulate in vivo environments and to enable high-content analysis without the use of animal models or humans. This is the benefit of many of these systems. On page 2, the

authors suggest that the lack of in vivo-like epithelial barrier tissue complexity is a limitation to these models. Again, can the authors provide specific information as to why this is and what information would be gained with ex vivo tissue explant models?

The authors again appreciate this feedback and have revised the manuscript to convey this point more clearly in the Introduction section. We agree that the benefit of existing tissue-on-a-chip/organ-on-a-chip models can recapitulate specific aspects of in vivo environments without the use of animal/human live tissue. However, they have largely been restricted to epithelial cell functions. The advantage that ex vivo models have over these is that they include the full cellular diversity and complexity. With a more physiologically relevant model researchers using this device can perform more accurate and realistic experiments. It also allows them to study the effects on other aspects of the tissue other than cells, such as the muscle layer, microbiome, neurons, and immune cells.

4. The techniques described here may not translate well to a scalable high-throughput system. Also, tissue explants can only survive for 72 hours in the current system. Authors should provide more experimental data to clarify the potential use of their system as a high-throughput live tissue assessment platform.

The authors would like to clarify that the reference to the high-throughput use of the system was meant to compare to the existing Ussing chamber systems and organ-on-a-chip systems where one organ tissue is interrogated at a time. Our current design allows up to three organ tissues to be interrogated at a time. Because the system architecture is scalable, the current design can be further scaled up to allow more organ tissues to be measured at the same time. However, due to the complexity of supporting subsystems, such as media pumps, we would agree with the reviewer that scaling up to the degree of throughput based on well plates, such as Transwell systems, would be challenging. We have modified the paper at the end of the Results section for clarification.

5. The bulk of the results section focuses on TEER system set up. Discussion of the biological results (especially those from intestinal tissues) is often phenomenological and contains technical details that would be more appropriate for supporting information. More space should be dedicated to discussing the physiological relevance and significance of their approaches and major findings from the experimental studies.

The major focus of this study was the design and execution of a microphysiological device that could assess TEER for multi-layered tissue ex vivo. Therefore, details of the TEER system are essential. The physiological experiments were conducted to determine whether the TEER measurements reflected real changes in the tissue. We used tissue manipulations that were previously characterized to deliberately degrade mucosal integrity (e.g., collagenase, acid) versus muscle integrity (e.g., cut). The vast majority of prior microphysiological studies have used epithelial cell monolayers. The results in the current study demonstrate that intestinal wall TEER (and thereby paracellular integrity) is not a simple function of the luminal epithelial monolayer. The experiments indicate the utility of this device design to assess changes in critical tissue function for future experiments.

Reviewers Comments #3

First, in the Introduction section, the authors claim that their approach allows for the long-term culture of ex vivo tissues compared with existing methods. However, the authors did not provide sufficient data or rationale to explain why this happens with their approach. While the application of microfluidics may be an enabling factor, additional supporting data are needed. For example, could you provide and compare tissue viability tests for microfluidic and static cultures in the proposed device?

The authors appreciate this feedback and revisions have been made in the Introduction section to address this issue. The revised text focuses on two main factors contributing to the long-term culture of ex-vivo tissue: microfluidics, and proper media composition. These two methods have proven to provide higher viability for ex-vivo tissues in several papers cited in the manuscript. Briefly, microfluidics provides fresh media in the form of laminar flow to the tissue throughout the experiment. This provides the sample with fresh nutrients and a more physically accurate environment (e.g., cyclic strain, fluid shear stress, and mechanical stretching). Microfluidics have also been shown in previous studies to increase viability compared to static media devices, such as Ussing Chambers (Eslami Amirabadi, et al., Ghiselli et al.). Furthermore, the media composition is just as important because it contains an oxygen gradient, a prebiotic (to maintain the natural microbiome), and a normoglycemic environment. We believe that the cited work and the physical confirmation of tissue viability through several biological markers is sufficient data to prove the efficacy of long-term viability in the proposed device.

Second, regarding the section of tissue viability, I have a couple of suggestions. This section is important as it validates the efficacy of the approach, especially for the long-term culture of ex vivo tissues. However, the authors have provided only qualitative analyses of animal colon epithelium three days post-culture, without any comparison to control groups. Please consider comparing the post-three-day tissues with the initial ex vivo epithelial tissues that are uncultured in the devices.

The authors appreciate this suggestion, we have updated Figure 5 with images showing tissue after 72h and tissue never cultured in a device. This provides evidence of tissue viability in the device after 72h. The updated figure shows a side-by-side comparison of claudin-1 immunoreactivity to highlight tight junction maintenance and UEA-1⁺ staining to highlight goblet cell preservation. These images together with quantitative data showing the number of goblet cells in each tissue (in vivo and ex vivo) demonstrate healthy tissue preservation ex vivo.

Additionally, alongside studies on tight junction degradation and goblet cell functions, could you also provide data on actual epithelial cell viability? This would directly demonstrate the ability of this approach to maintain the viability of ex vivo epithelial cells.

Thank you for this suggestion. We agree that providing data on epithelial cell viability would further support our findings. We have provided quantitative analysis of goblet cell (specialized epithelial cells) number between in vivo and 72 h ex vivo tissue which has been added in figure 5a(vi). We chose to analyze goblet cell numbers because a decrease in goblet cell number is hallmark to tissue damage in several intestinal disorders. There was no significant difference in

goblet cell number between ex vivo and in vivo tissue indicating that 72 h ex vivo tissue had similar epithelial cell viability compared to in vivo tissue.

Regarding Figure 5v, could you provide the temporal changes in TEER at 0, 24, 48, and 72 hours for the various media compositions, instead of only showing changes between 24 and 48 hours? This additional data would validate the efficacy and reliability of this approach by demonstrating whether all sample tissues start with similar TEER values and confirming that this approach can function effectively for 72 hours.

Thank you for this comment, this is an important point in how we interpreted the data. We updated the Method section to better explain the reasoning for the data shown. There are several reasons for showing only 24-48 h in the TEER experiment for Figure 5v. All experiments were stopped at 48 h to keep consistency for all tissues. Tissue that was treated with acidic media suffered sufficient degradation that continuing onto 72 h would be uninformative. Since the acidic media can only be studied up to 48h, then the other two media compositions were done the same to keep everything consistent. The TEER was also not considered in the first 24 h because we used this time as a calibration/settling period for the tissue. During the settling period, we found that TEER values varied significantly to adapt to the new environment, and the TEER values settled by around 24h. These settled TEER values were considered as the “starting” TEER values and were compared to the ending (48h) TEER values to find the change in TEER over time. We believe that the appropriate test of efficacy and reliability of the system for 72h comes from the viability tests described in the *Tissue Viability* portion of the Results section. The viability of the tissue is addressed through various biological health markers.

I am also concerned about the biocompatibility of the Anycubics UV resin used to construct the microfluidic chamber. While the authors attempted to mitigate any harmful effects by ensuring the resin is fully cured, I believe that the biocompatibility of the resin still remains uncertain. To address this issue, I suggest conducting a straightforward experiment comparing cell viability between tissues (or any type cells) cultured in wells made from the proposed resin and those cultured in commercially available well plates.

The authors appreciate this comment and provide the suggested assessments in the context of the device and tissue of study. We added images of goblet cells (UEA-1) and claudin-1 of tissue that never went into the device (in vivo). In vivo and 72 h ex vivo tissue have comparable goblet cell and claudin-1 morphology. We believe that the evidence of tissue viability through biological health markers up to 72h is sufficient to indicate biocompatibility of the chambers in the context of the current study.

In the section discussing the proximal vs. distal colon, the difference in TEER between these colon segments is attributed to their in vivo pH differences. However, it appears that the same pH level of the tissue culture media and their microfluidic flowing were used for both ex vivo tissues in the experimental setup suggesting the pH conditions for both tissue sections might be identical. Therefore, further discussion on this aspect is critical to assess the reliability of the proposed approach.

The authors greatly appreciate this comment and after careful consideration we concur. The pH differences in vivo were not translated to our experiment setup ex vivo. For this reason, we have removed the pH explanation. The main reason for including this data is because it is consistent with previous findings in mouse colon tissue (Thomson et al.). The experiments did not address biological causes for the differences between regions.

Communications Engineering, Nature Publishing

Response to Reviewers #2:

Reviewer #2:

Comments on Highlights of System Performance:

The authors highlight the current system as a high-throughput system due to its capabilities of being a microphysiological system. However, they do not demonstrate this capability in the manuscript. A 3X throughput doesn't sound high throughput in the context of the microfluidic or organ-on-a-chip research field. I suggest that this section should focus on the system's ability to measure TEER values ex vivo and discuss why this has advantages over previous TEER measurement systems.

Thank you for your comment. We agree with the reviewer about the use of words, “high throughput” in the paper. We have removed this phrase from the paper to avoid potential misunderstandings from readers. We instead only highlighted our capability of providing multiple simultaneous measurements throughout an experiment, because many existing devices targeting ex vivo tissue’s over long-term experiments are capable of only one tissue at a time. Per reviewer’s request, we have also updated the “Highlights of System Performance” Section in the manuscript to better explain the advantages of our system through attributes such as TEER measurement capability, longer viability, and others, when compared with existing epithelial transport systems listed in Table 2.